# News-Driven Stock Prediction Using Noisy Equity State Representation

## Abstract

News-driven stock prediction investigates the correlation between news events and stock price movements. Previous work has considered effective ways for representing news events and their sequences, but rarely exploited the representation of underlying equity states. We address this issue by making use of a recurrent neural network to represent an equity state transition sequence, integrating news representation using contextualized embeddings as inputs to the state transition mechanism. Thanks to the separation of news and equity representations, our model can accommodate additional input factors. We design a novel random noise factor for modeling influencing factors beyond news events, and a future event factor to address the delay of news information (e.g., insider trading). Results show that the proposed model outperforms strong baselines in the literature.

## 1 Introduction

Stock movement prediction (Ding et al., 2014; 2015) is a central task in computational quantitative finance. With recent advances in deep learning and natural language processing (NLP), event-driven stock prediction has received increasing research attention (Xie et al., 2013; Ding et al., 2015). The goal is to predict the movement of stock prices according to financial news. Previous work adopts a relatively simple model on the stock movement process, casting price change as a response to a set of news. The prediction model can therefore be viewed as variation of a classifier that takes news as input and yields a movement direction output. Investigations have focused on news representation, where bag-of-words (Kogan et al., 2009), named entities (Schumaker & Chen, 2009), event structures (Ding et al., 2014) or neural representation features (Ding et al., 2015; Xu & Cohen, 2018) are considered.

Intuitively, news events carry information on important changes of company management, market, revenue and other factors, which can affect the fundamental values of equities, and thereby can consequently impact the stock price, as shown in Figure 1. Properly representing news events is key to modeling such impact on the market. However, the stock market movement can also be influenced by accumulated effects of fundamental changes over time, the overall market sentiment, and other volatile factors, which can be considered as noise to analytical models. These factors have been relatively less modeled by existing work on event-driven stock prediction. For example, although there has been work modeling long-term event impacts by representing event sequences (Ding et al., 2015), little work has considered representing fundamental values directly.

To address these issues, we consider representing the equity state directly using a recurrent neural network over time and propose the stock movement prediction network using Noisy Equity State representation (NES). At each time step, the equity state reflects the current stock price trend, and can be used directly for predicting the next movement. The advantage of separating news representation from equity state representation is that factors beyond news can be modeled as additional input in the recurrent state transition process. Although such factors can be calculated using external tools such as sentiment classification over tweet data, we simply treat them as a random noise factor. The reason is two-fold. First, in practice, noise is inevitable in stock prediction and no single mathematical model can perfectly fit the stock price movement distribution. Second, for fair comparison with existing work on news-driven stock prediction, no additional input should be used on top of standard benchmark input settings.

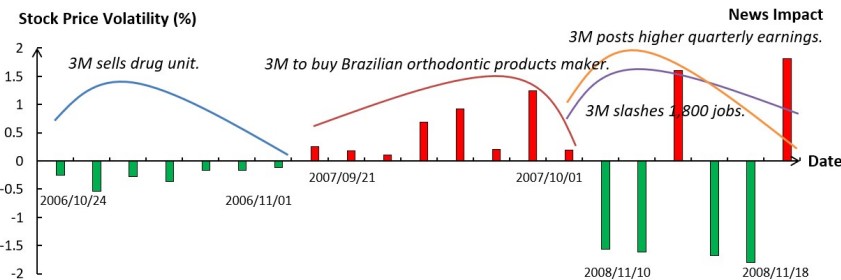

Figure 1: The stock price volatility and the news impact on *3M Company*. Over the first and the second periods, there was only one event. And in the third period, there were two events affecting the stock price movements simultaneously.

The input to each recurrent equity state transition consists of a news factor and a noise factor. The news factor consists of three components, namely past news within thrity days, present news within a trading day, and future news within seven trading days. We use real historical news for the future news component for training, and predicted future news according to the current equity state during testing. The motivation for modeling future news is to address the negative effect of delayed news release and insider trading on the prediction accuracy. We represent each news event using contextualized representation of the news title, and aggregate news representations by using the current equity state as a query to conduct attention. The noise factor is integrated into the model by using the current equity state to derive a normal noise distribution specific to the trading day, and then sampling a noise vector.

Compared with existing work, our model has three main potential advantages. First, the relative importance of news can be easily visualized using attention. Second, insider trading effect is explicitly handled. Third, noise effect is integrated into the model. All three benefit result directly from the direct representation of equity state. Experiments over the benchmark of Ding et al. (2015) show that our method outperforms strong baselines, giving the best reported results in the literature. To our knowledge, we are the first to explicitly model both events and fundamental equity states for news-driven stock movement prediction, and the first to consider noise factors using a neural random sample module.

## 2    RELATED WORK

**Modeling Price Movements Correlation** Most existing work on event-driven stock prediction treats the representation of news events independently using bag-of-words (Kogan et al., 2009), named entities (Schumaker & Chen, 2009), semantic frames (Xie et al., 2013), event structures (Ding et al., 2014), event embeddings (Ding et al., 2015) or knowledge bases (Ding et al., 2016). In contrast, work on time-series based stock prediction (Levine & Zervos, 1996; Amihud, 2002; Xu & Cohen, 2018; Zhang et al., 2018a) aims to capture continuous movements of prices themselves. There has also been work modeling the correlations between samples by sparse matrix factorization (Wong et al., 2014), hidden Markov model (Zhang et al., 2018a) and Bi-RNNs (Xu & Cohen, 2018; Yang et al., 2019) using both news and historical price data. Some work models the correlations among different stocks by pre-defined correlation graph (Peng & Jiang, 2016) and tensor factorization (Zhang et al., 2018b). Our work is different in that we use only news events as inputs, and our recurrent states are additionally designed to accommodate noise.

**Explainable Prediction** Rationalization is an important problem for news-driven stock price movement prediction, which is to find the most important news event along with the model's prediction. Factorization, such as sparse matrix factorization (Wong et al., 2014) and tensor factorization (Zhang et al., 2018b), is a popular method where results can be traced back upon the input features. Our attention-based module achieves a similar goal yet has linear time complexity on feature size. Chang et al. (2016) use attention for characterizing the influence of individual news within a trading day for predicting the cumulative abnormal return in a three-day window. Yang et al. (2019) apply dual-layer attention to predict the stock movement by using news published in the previous six days. Each day's news embeddings and seven days' embeddings are summed by the layer. Our work is

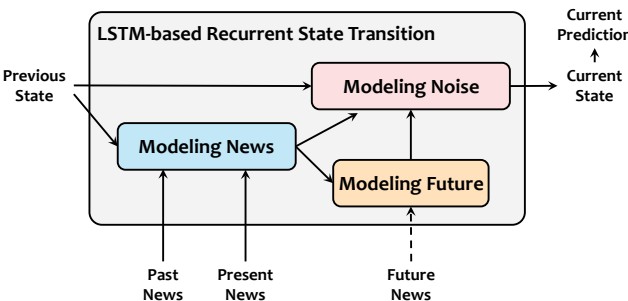

Figure 2: The model framework of NES representing a trading day in the trading sequence. The solid lines are used both in the training and the evaluating procedures, while the dotted line is only used in the training procedure.

different from Yang et al. (2019) in that our news events attention is query-based, which is more strongly related to noisy recurrent states. In contrast, their attention is not query-based and tends to output the same result for each day even if the previous day's decision is changed.

## 3 METHOD

Following previous work (Ding et al., 2014; 2015), the task is defined as a binary classification task. Formally, given a trading day $x$, the input is a history news set about a targeted stock or index and the output is a label $y \in \{+1, -1\}$ indicating whether the adjusted closing price[1] $p_x$ will be greater than $p_{x-1}$ ($y = +1$) or not ($y = -1$).

The structure of our model is shown in Figure 2. The equity state is modeled using LSTM (Hochreiter & Schmidhuber, 1997), which serves as a basis for making prediction. News events are separately represented as inputs to the state transition process. For each trading day, we consider the news events in the day as well as the past news events using neural attention (Vaswani et al., 2017). Considering the impacts of insider trading, we also involve future news in the training procedure. To model the high stochasticity of stock markets, we sample an additive noise using a neural module.

Considering the general principle of sample independence, building temporal connections between individual trading days in the training set can harm training (Xu & Cohen, 2018) and we find it easy to overfit. To address this issue, we extended the time span of one sample to $T$ previous continuous trading days ($t - T + 1, t - T + 2, ..., t - 1, t$), which we call a trading sequence. Trading sequences are used as the basic training elements in this paper.

### 3.1 LSTM-BASED RECURRENT STATE TRANSITION

Formally, we denote the equity state using $z$. In each trading day, the recurrent state transition can be written as

$$z_t = f(z'_t), \ z'_t = \overrightarrow{\text{LSTM}}(v_t, z_{t-1}) \tag{1}$$

where $v_t$ represents the news event on the trading day $t$ and $f$ is a function for the noise effect. Non-linear compositional effects of multiple events can also be captured in a time window because of the LSTM representation. To predict the current day stock price movement, we use the sequential state $z_t$ to make binary classification

$$\hat{p}_t = \text{softmax}(W^y z_t), \ \hat{y}_t = \underset{i \in \{+1, -1\}}{\arg\max} \ \hat{p}_t(\hat{y}_t = i | t) \tag{2}$$

where $\hat{p}_t$ is the estimated probabilities for up ($+1$) and down ($-1$) movements, $\hat{y}_t$ is the predicted label, $W^y$ is the classification parameter and $t$ is the input trading day.

---

[1] https://yhoo.it/3i5UkRh

## 3.2 Modeling News Events

We separately represent long-term and short-term impact of news events for each day $t$ in a trading sequence. For short-term impact, we use news articles published after the previous trading day $t-1$ and before the trading day $t$ as the set of present news. Similarly, for long-term impact, we use news articles within thirty calendar days as the set of past news.

Each news article is simply regarded as a separate news event. We extract the headline and use ELMo (Peters et al., 2018) to transform it to $V$-dim hidden state by concatenating the bidirectional output hidden states of the first and the last words as the basic representation of a news event. By stacking the vector representations of different news articles, we obtain two embedding matrices $\boldsymbol{C}'_t$ and $\boldsymbol{B}'_t$ for the present and past news events, respectively, as

$$\boldsymbol{e} = [\overrightarrow{\text{ELMo}}(\boldsymbol{h})^{-1}, \overleftarrow{\text{ELMo}}(\boldsymbol{h})^0] \tag{3}$$

$$\boldsymbol{C}'_t = [\boldsymbol{e}^1_c \oplus \boldsymbol{e}^2_c \oplus ... \oplus \boldsymbol{e}^{l_c}_c] \,,\; \boldsymbol{B}'_t = [\boldsymbol{e}^1_b \oplus \boldsymbol{e}^2_b \oplus ... \oplus \boldsymbol{e}^{l_b}_b] \tag{4}$$

where $[\,,\,]$ is the vector concatenation operation, $[\,\oplus\,]$ is the vector stacking operation, $l_c$ is the size of present news set and $l_b$ is the size of past news set.

To make the model more numerically stable and avoid overfitting, we apply the over-parameterized method of Merity (2019) to the news event embedding matrices

$$\boldsymbol{C}_t = \sigma(\boldsymbol{W}^f \boldsymbol{C}'_t) \odot \tanh(\boldsymbol{W}^c \boldsymbol{C}'_t) \,,\; \boldsymbol{B}_t = \sigma(\boldsymbol{W}^f \boldsymbol{B}'_t) \odot \tanh(\boldsymbol{W}^c \boldsymbol{B}'_t) \tag{5}$$

where $\odot$ is element-wise multiplication, $\sigma(\cdot)$ is the sigmoid function, and $\boldsymbol{W}^f$ and $\boldsymbol{W}^c$ are model parameters.

Due to the unequal importance fact of news events with regard to the stock price movement in day $t$, we use scaled dot-product attention (Vaswani et al., 2017) to capture the influence of news $\boldsymbol{C}_t$ and $\boldsymbol{B}_t$ to the current day stock movement. Formally, we first transform the last trading day's equity state $\boldsymbol{z}_{t-1}$ to a query vector $\boldsymbol{q}_t$, and then calculate two attention score vectors $\boldsymbol{\gamma}_t$ and $\boldsymbol{\beta}_t$ for the present and past news events, respectively as

$$\boldsymbol{q}_t = \tanh(\boldsymbol{W}^q \boldsymbol{z}_{t-1}) \,,\; \boldsymbol{\gamma}_t = \text{softmax}(\frac{\boldsymbol{C}^i_t \boldsymbol{q}_t}{\sqrt{V}}) \,,\; \boldsymbol{\beta}_t = \text{softmax}(\frac{\boldsymbol{B}^i_t \boldsymbol{q}_t}{\sqrt{V}}) \tag{6}$$

We sum the news event embedding matrices to obtain two final news representation vectors $\boldsymbol{c}_t$ and $\boldsymbol{b}_t$ on the trading day $t$ according to the weights $\boldsymbol{\gamma}_t$ and $\boldsymbol{\beta}_t$, respectively, as

$$\boldsymbol{c}_t = \tanh(\sum_{i=1}^{l_c} \boldsymbol{\gamma}^i_t \boldsymbol{C}^i_t) \,,\; \boldsymbol{b}_t = \tanh(\sum_{i=1}^{l_b} \boldsymbol{\beta}^i_t \boldsymbol{B}^i_t) \tag{7}$$

## 3.3 Modeling Future News

We find that news events can exert an influence on the stock price movement *before* being released, which can be attributed to news delay or insider trading (Seyhun, 1992) factors. To this end, we propose a novel future event module to represent backward news influence. In this paper, we define future news events as those that are published within seven calendar days after the trading day $t$.

Similarly to the past and present news events, we stack the ELMo embeddings (Peters et al., 2018) of future news event headlines to an embedding matrix $\boldsymbol{A}'_t$, and then adapt the over-parameterized method and sum the stacked embedding vectors by scaled dot-product attention. Formally, the future news events impact vector $\boldsymbol{a}_t$ on the trading day $t$ is calculated as

$$\boldsymbol{A}_t = \sigma(\boldsymbol{W}^f \boldsymbol{A}'_t) \odot \tanh(\boldsymbol{W}^c \boldsymbol{A}'_t) \,,\; \boldsymbol{\alpha}_t = \text{softmax}(\frac{\boldsymbol{A}^i_t \boldsymbol{q}_t}{\sqrt{V}}) \,,\; \boldsymbol{a}_t = \tanh(\sum_{i=1}^{l_a} \boldsymbol{\alpha}^i_t \boldsymbol{A}^i_t) \tag{8}$$

where $l_a$ is the size of future news set.

The above steps can be used to reduce overfitting in the training procedure, where the future event module is trained over gold "future" data over historical events. At test time, future news events are

not accessible. To address this issue, we use a non-linear transformation to estimate a future news events impact vector $\hat{a}_t$ with the past and present news events impact vectors $b_t$ and $c_t$ as

$$\hat{a}_t = \tanh(\boldsymbol{W}^a[\boldsymbol{c}_t, \boldsymbol{b}_t]) \tag{9}$$

where $[\ ,\ ]$ is the vector concatenation operation and $\boldsymbol{W}^a$ is model parameter. We concatenate the afore-mentioned three types of news event vectors $\boldsymbol{a}_t$, $\boldsymbol{b}_t$ and $\boldsymbol{c}_t$ to obtain a final news event input $\boldsymbol{v}_t$ for the recurrent state transition on trading day $t$.

### 3.4 MODELING NOISE

All factors beyond input news articles, such as sentiments, expectations and noise are explicitly modeled as noise using a random factor. For each trading day, we sample a random factor from a normal distribution $\mathcal{N}(\boldsymbol{0}, \boldsymbol{\sigma}_t)$ parameterized by $\boldsymbol{z}'_t$ as

$$\boldsymbol{\sigma}_t = \sqrt{\exp(\tanh(\boldsymbol{W}^\sigma \boldsymbol{z}'_t))} \tag{10}$$

In addition, to facilitate back-propagation in training, we use re-parameterization for normal distributions (Srivastava & Sutton, 2017), drawing a sample random factor from a parameter-free distribution to obtain the noisy recurrent state $\boldsymbol{z}_t$ as

$$\boldsymbol{z}_t = \tanh(\boldsymbol{z}'_t + \boldsymbol{\sigma}_t \boldsymbol{\epsilon}_t) \ , \ \boldsymbol{\epsilon}_t \sim \mathcal{N}(\boldsymbol{0}, \boldsymbol{1}) \tag{11}$$

### 3.5 TRAINING OBJECTIVE

For training, we consider two main terms for defining the loss function. The first term is a cross entropy loss for the stock movement probabilities $\hat{p}_t$ and gold labels $y_t$, and the second term is the mean squared error between the estimated future impact vector $\hat{a}_t$ and the true future impact vector $\boldsymbol{a}_t$ stated in Section 3.3. Formally, the loss for a trading sequence containing $T$ trading days with standard $L_2$ regularization is calculated as

$$L_{ce} = \sum_{t=1}^{T} -\log(1 - \hat{p}_t(y_t|t)) \ , \ L_{mse} = \frac{1}{V} \sum_{t=1}^{T} \sum_{i=1}^{V} (\hat{a}_t^i - a_t^i)^2 \tag{12}$$

$$L_{total} = L_{ce} + \theta L_{mse} + \lambda \|\Phi\|_2^2 \tag{13}$$

where $\theta$ is a hyper-parameter which indicates the relative importance of $L_{mse}$ to $L_{ce}$, $\Phi$ is the set of trainable parameters in the model and $\lambda$ is the regularization weight.

## 4 EXPERIMENTS

We use the public financial news dataset released by Ding et al. (2014), which is crawled from Reuters and Bloomberg over the period from October 2006 to November 2013, and follow their method for splitting the dataset. Experiments are conducted on predicting the Standard & Poor's 500 stock (S&P 500) index and a set of selected individual stocks, obtaining prices from Yahoo Finance[2]. We report the final results after tuning hyper-parameters in the development experiments.

### 4.1 SETTINGS

The hyper-parameters are shown in Table 1. We use mini-batch and SGD with momentum to update the parameters. Most of the hyper-parameters are chosen according to development experiments, while the dropout rate $r$ and SGD momentum $\mu$ are set according to common values. Following previous work (Xie et al., 2013; Ding et al., 2014; Xu & Cohen, 2018), we adopt accuracy and Matthews Correlation Coefficient (MCC) to evaluate model performances. MCC is applied because it avoids bias due to data skew.

### 4.2 DEVELOPMENT EXPERIMENTS

### 4.2.1 INITIALIZING NOISY RECURRENT STATES

---

[2] https://finance.yahoo.com/

Table 1: Hyper-parameters setting.

| Name | Value |
|---|---|
| batch size | 16 |
| learning rate $lr$ | 0.005 |
| SGD momentum $\mu$ | 0.9 |
| dropout rate $r$ | 0.3 |
| MSE loss weight $\theta$ | 0.4 |
| regularization weight $\lambda$ | 0.0005 |
| news embedding dimension $V$ | 256 |
| recurrent state dimension $D$ | 100 |

Figure 3: Results of different $T$.

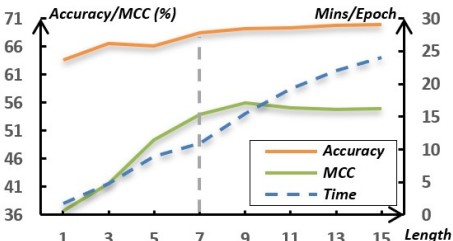

As the first set of development experiments, we compare different ways to initialize the noisy equity states of NES. For each trading day, we compare the results whether states transitions are modeled or not as well as the initialization methods of the equity states. *NES_Sing_R* is randomly initializing the states for each single trading day. *NES_Sing_Z* is initializing the states as zeros for each single trading day. *NES_Seq_R* is randomly initializing the first states for each trading sequence only. *NES_Seq_Z* is initializing the first states as zeros for each trading sequence only.

Table 2: Results on initializing the noisy equity states.

| | Accuracy | MCC |
|---|---|---|
| NES_Sing_R | 62.91% | 0.3704 |
| NES_Sing_Z | 63.63% | 0.3672 |
| NES_Seq_R | 67.94% | 0.5141 |
| NES_Seq_Z | **68.51%** | **0.5392** |

The random initialization method returns a tensor filled with random numbers from the standard normal distribution $\mathcal{N}(\mathbf{0}, \mathbf{1})$. The results on the S&P 500 index are shown in Table 2. We can see that modeling recurrent sequences (*NES_Seq_R* and *NES_Seq_Z*) gives better result than treating each trading day separately (*NES_Sing_R* and *NES_Sing_Z*), which shows that modeling equity states can capture long-term and non-linear compositional effects of multiple events. From another perspective, comparision with the models *NES_Sing_R* and *NES_Sing_Z* also demonstrates the strengths of our basic representations of news events in isolation. In particular, we can also see that using only the basic news events representations is not sufficient for index prediction, while a combination with the equity state transition can give better results. This is consistent with previous reports on news-driven stock prediction (Ding et al., 2015). By comparing the results of *NES_Seq_R* and *NES_Seq_Z*, we decide to use zero initialization in the remaining experiments.

### 4.2.2 STUDY ON TRADING SEQUENCE LENGTH

We use the development set to find a suitable length $T$ for a trading sequence, searched from $\{1, 3, 5, 7, 9, 11, 13, 15\}$. The S&P 500 index prediction results and time per training epoch on the development set are shown in Figure 3. We can see that the accuracy and MCC are positively correlated with $T$. When $T \geq 7$, the growth of MCC becomes slower, while the running time per training epoch is nearly linear with respect to $T$. We therefore choose the hyper-parameter $T = 7$ for the remaining experiments.

### 4.3 MAIN RESULTS

We compare our approach with five strong baselines on predicting the S&P 500 index. Luss & D'Aspremont (2015) use bag-of-words to represent news documents and adopt Support Vector Machines. Ding et al. (2015) use event embeddings as input and convolutional neural network to model a sequence of events. Ding et al. (2016) empower event embeddings with knowledge bases like YAGO and also adopt convolutional neural networks as the event sequence model. dos Santos Pinheiro & Dras (2017)

Table 3: Test set results on predicting S&P 500 index.

| | Accuracy | MCC |
|---|---|---|
| Luss & D'Aspremont (2015) | 56.38% | 0.0711 |
| Ding et al. (2015) | 64.21% | 0.4035 |
| Ding et al. (2016) | 66.93% | 0.5072 |
| dos Santos Pinheiro & Dras (2017) | 63.34% | - |
| Lin et al. (2017) | 64.55% | - |
| **NES** | **67.34%** | **0.5475** |

Table 5: Test set results of individual stock price movement prediction.

| Stock | Sector | Company News | | | Sector News | | | All News | |
|-------|--------|------|----------|-----|------|----------|-----|----------|-----|
| | | #docs | Accuracy | MCC | #docs | Accuracy | MCC | Accuracy | MCC |
| Apple | IT | 2,398 | 69.21% | 0.5632 | 12,812 | 64.35% | 0.3861 | 56.14% | 0.2355 |
| Citigroup | Financials | 2,058 | 63.57% | 0.5193 | 117,659 | 56.29% | 0.3021 | 55.15% | 0.1852 |
| Boeing Company | Industrials | 1,870 | 66.25% | 0.4423 | 17,969 | 61.35% | 0.2719 | 57.23% | 0.1824 |
| Google | Communication | 1,762 | 66.13% | 0.3717 | 13,344 | 60.47% | 0.2644 | 58.41% | 0.1387 |
| Wells Fargo | Financials | 845 | 61.64% | 0.3944 | 117,659 | 57.34% | 0.1294 | 54.64% | 0.0823 |

use a fully connected model and character-level embedding with LSTM to encode news sequences. Lin et al. (2017) use recurrent neural networks with skip-thought vectors to represent news text.

Table 3 shows the test set results. All the models make prediction based on news input only. From the table we can see that NES achieves the best results on the test set, which demonstrates the advantage of separately representing equity state sequences and integrating noise features. By comparing with Luss & D'Aspremont (2015), we can find that using news event embeddings and equity state representations can be more effective and also flexible when dealing with high-dimension features. When comparing with Ding et al. (2015) and the knowledge-enhanced method (Ding et al., 2016), we find that modeling the correlations between trading days in trading sequences can better capture the compositional effects of multiple news events. In addition, by comparing with dos Santos Pinheiro & Dras (2017) and Lin et al. (2017), we also find that modeling the noise by using a state-related random factor is very beneficial and effective to meet the aim to deal with the high market stochasticity.

## 4.4 ABLATION STUDY ON NEWS AND NOISE

We explore the effects of news events and random noise with ablation on the test set. By ablating the past news, the present news, future news and the noise factor, respectively, the S&P 500 index prediction results of the ablated models are shown in Table 4. First, without using the past news events as input, the result becomes the lowest. This shows that history news articles contain the largest amount of relevant news events. In addition, since we set the trading sequence length $T = 7$ and we only use the news between two trading days as the present news, those ablated news which was published before the date range will not be involved in our model, while the ablated present or past news will be a part of the input on adjacent trading days.

Table 4: Results of ablation study.

| | Accuracy | MCC |
|-------|----------|-----|
| **NES** | **67.34%** | **0.5475** |
| - Past News | 62.17% | 0.4421 |
| - Present News | 64.73% | 0.4823 |
| - Future News | 64.58% | 0.4781 |
| - Noise | 63.90% | 0.4608 |

Second, we observe that using future news events is more effective than using the present news events. On the one hand, it confirms the importances to involve the future news in NES, which can deal with insider trading factors to some extent. On the other hand, the results also benefits from the recurrent state transition nature of the model, as the future news impact on the $(t − 1)$-th day can be carried forward to the $t$-th day in the equity state to compensate absence of the present news events.

The effect of the noise factor is lower only to modeling the past news events, but higher than the other ablated models, which demonstrates the effectiveness of the noise factor module. This shows the advantage of considering factors beyond news events. It also directly benefits from our representation of equity states, which gives a basis for calculating a noise distribution conditioned on accumulated history.

## 4.5 PREDICTING INDIVIDUAL STOCK MOVEMENTS

Other than predicting the S&P 500 index, we also investigate the effectiveness of our approach on individual stock prediction using the test set. We count the amounts of individual company related news events for each company by name matching, and select five well known companies from four different sectors with the most news, *Apple*, *Citigroup*, *Boeing Company*, *Google* and *Wells Fargo*. The sectors are decided by the Global Industry Classification Standard. For each company, we

| No. | Date | News Events |
|-----|------|-------------|
| 1 | 2013-06-17 | Apple Joins Facebook, Microsoft in Outlining Data Requests |
| 2 | 2013-06-21 | Apple Wins Suit Against Samsung in Japan on Screen Effects |
| 3 | 2013-06-24 | Apple Falls Below $400 Amid IPhone Slump, Worker Exits |
| 4 | 2013-06-25 | Samsung Beats Apple in Japanese Patent Suit on Syncing |
| 5 | 2013-07-02 | Apple plans Nevada solar farm in clean energy push for data centers |
| 6 | 2013-07-10 | Apple Faces Damages Trial Over E-Book Antitrust Violation |
| 7 | 2013-07-17 | Apple May Delay Introduction of IPhone 5S, Commercial Times Says |
| 8 | 2013-07-19 | Apple Said to Buy HopStop, Pushing Deeper Into Maps |
| 9 | 2013-07-21 | High-End Smartphone Boom Ending as Price Drop Hits Apple |
| 10 | 2013-07-22 | Apple Developer Website Taken Down After Hacker Attack |

Attention heat map (rows correspond to News Events 1–10; columns 2013-07-15, 2013-07-16, 2013-07-17, 2013-07-18, 2013-07-19, 2013-07-22, 2013-07-23):

| 2013-07-15 | 2013-07-16 | 2013-07-17 | 2013-07-18 | 2013-07-19 | 2013-07-22 | 2013-07-23 |
|------------|------------|------------|------------|------------|------------|------------|
| 0.094 | 0.024 | 0.061 | 0.052 | 0 | 0 | 0 |
| 0.064 | 0.053 | 0.086 | 0.042 | 0.086 | 0.033 | 0 |
| 0.0064 | 0.087 | 0.053 | 0.078 | 0.021 | 0.084 | 0.041 |
| 0.055 | 0.083 | 0.048 | 0.018 | 0.026 | 0.025 | 0.084 |
| 0.074 | 0.079 | 0.029 | 0.051 | 0.042 | 0.034 | 0.078 |
| 0.045 | 0.043 | 0.034 | 0.058 | 0.0047 | 0.089 | 0.067 |
| 0 | 0 | 0 | 0.1 | 0.08 | 0.091 | 0.016 |
| 0 | 0 | 0 | 0 | 0 | 0.044 | 0.045 |
| 0 | 0 | 0 | 0 | 0 | 0 | 0.011 |
| 0 | 0 | 0 | 0 | 0 | 0 | 0.098 |
| **Gold** 1 | 1 | 1 | 1 | -1 | 1 | -1 |
| **Pred** 1 | 1 | 1 | -1 | 1 | 1 | -1 |

Figure 4: Attention visualization and test set results comparison of the trading sequence [07/15/2013, 07/23/2013] when predicting Apple Inc.'s stock price movements using only company news.

prepare not only news events about itself, but also news events about the whole sector. We use company news, sector news and all financial news to predict individual stock price movements, respectively. The experimental results and news statistics are listed in Table 5.

Individual stock prediction by only using company news dramatically outperforms that using sector news and all news, which demonstrates a negative correlation between total used amounts of news events and model performance. The main reason may be that company-related news events can more directly affect the volatility of company shares, while sector news and all news contain many irrelevant news events, which obstruct NES's learning the underlaying stock price movement trends.

### 4.6 CASE STUDY

To look into what news event contributes the most to our prediction result, we further analyze the test set results of the model for *Apple Inc.*'s stock price movements only using company news, which achieves the best results among the five selected companies mentioned before. As shown in Figure 4, we take the trading sequence from 07/15/2013 to 07/23/2013 for illustration. The table on the left shows the top-ten news events, while attention visualization and results are shown on the right chart. The news events listed on the left table are ranked by the attention scores from the past news events, which are the most effective news according to the ablation study. There are some zeros in the attention heat map because these news do not belong to the corresponding trading days.

We can find that the News Event 1 has been correlated with the stock price rises on 07/15/2013, but for the next two trading days, its impact fades out. On 07/18/2013, the News Event 7 begins to show its impact. However, NES pays more attention to it compared with other events, which leads to the incorrect prediction that the stock price decreases. On the next trading day, our model infers that the impact of the News Event 2 is bigger than that of the News Event 7, which leads to an incorrect prediction again. From these findings, we can see that NES tends to pay more attention to a new event when it first occurs, which offers us a potential improving direction in the future.

### 5 CONCLUSION

We investigated explicit modeling of equity state sequences in news-driven stock prediction by using an LSTM to model the recurrent state transition, adding news impact and noise impact by using attention and noise sampling, respectively. Results show that our method is highly effective, giving the best performance on a standard benchmark for stock index movement prediction. To our knowledge, we are the first to explicitly model both events and noise factors for news-driven stock movement prediction.

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
