# OpenReview forum: "News-Driven Stock Prediction Using Noisy Equity State Representation"
_ICLR.cc/2021/Conference — Reject_

### Official Review · AnonReviewer2 · 2020-10-20
**Interesting paper, but the writing skills and effectiveness of proposed modules should be improved.**

**Rating:** 5
**Confidence:** 4

**Review:**

- Summary: A stock return prediction algorithm using news events is proposed. The authors claim that separately representing the news and equity states is a novel part of this paper. Also, the random noise factor and future event factors are proposed as novel modules.
- Quality: The paper is somewhat unclearly written and the performance improvement compared with conventional methods is marginal. The novelty is marginal or the authors should explain the novel contents in detail.
- Clarity: The paper contains some unclear explanations and inappropriate notations but could be improved.
  - Regarding notation, the authors need to be more strict.
  - Why the trading day is denoted by $x$ in Sec. 3 and denoted by $t$ in other text? It should be consistent.
   - Eq. (2). Why there is an 'i' on the upper side of argmax? If it is a typo, then please fix it.
   - Eq. (2). Is the t in the right hand side is necessary? (i.e. the $| t$ in the $\hat{p_t}(\hat{y_t} = i | t)$).
   - In the Equation (2) and (3), the $\overleftarrow{}$ and $\overrightarrow{}$ seems redundant representation. The authors could simplify the notation.
  - What is $h$ in Eq (3)? I can only guess but the authors should clarify.
   - There is no explicit definition of the equity state $z$. I guess that most of the ML people would easily figure out that it is an implicit state handled by LSTM, however, the people in the finance field may be confused that it could be an explanatory state (factor) such as momentum, volatility, or mean reversion. Also, the original news events are represented in natural language but it is represented as a news event vector $v_t$ in this paper. I suggest that the authors add an explicit explanation of the definition for a more self-contained paper.
   - Through subsections 3.2, 3.3, and 3.4, the submodules are explained with many notations. But it is difficult to figure out what the inputs and outputs are. I suggest the authors modify Figure 2 to include associated variables.
  - Equation (3)~(8) can be rearranged by operations since the past, present, and future news are computed in a similar way.
   - Sec. 3.2 The long-term articles may include short-term articles, and it seems redundant. Why not merge them into one module? (i.e. weighting scheme based on recent time).
  - Is the $v_t=[a_t, b_t, c_t]$ for training and $v_t=[\hat{a}_t, b_t, c_t]$ for test? It should be clarified in Sec. 3.3.
- Originality: I barely see originality in this paper.
- Significance: I barely see significant modules proposed in this paper.
- Pros
- Cons
   - Random noise factor.
      - One of the two novel features in this paper is the random noise factor. The authors claimed that it could model unknown factors such as sentiments, however, what I understand is that it is a simple data augmentation function. If it is an important module, the authors could assign more pages to explain and showing experimental results associated with the module.
   - Future news
    - What does it mean by predicting future news $\hat{a}_t$ using the past news $c_t$ and present news $b_t$ in Equation (9)? I don't make sense to me that future news can be predicted.
     - Throughout the paper, it should be very careful and clear about explaining future news. It may confuse the readers that the algorithm employs future data even in the test. Only careful readers would find out the "At test time, future news events are not accessible". It should be emphasized.
  - Ablation study
     - Excluding only one module significantly drops the predicted performance and could not improve than Ding et al.(2016). It means that the surviving modules may not be novel and effective in terms of performance.
- Experiments
   - I recommend the authors try other evaluation metrics such as the Sharpe ratio, to evaluate the realistic wealth curve. (i.e. if the model makes the right prediction for small movement and makes the wrong prediction for large price movement, then the wealth will eventually be reduced).
   - Ablation study is also performed properly.

---

### Official Review · AnonReviewer1 · 2020-10-25
**Presentation needs improvement**

**Rating:** 5
**Confidence:** 4

**Review:**

In this paper, the authors propose a model for integrating news representations for stock predictions. The authors claim that their proposed model outperforms the previous baselines.

Here are my comments:

1. Presentation needs improvement. Some statements are difficult to follow, or more explanations are needed. For example, "little work has considered representing fundamental values directly". What does "fundamental values" mean? This phase only occurs once in the introduction part. The authors do not elaborate on how to model fundamental values in their method. The benefits of modeling fundamental values should be illustrated.

2. Following the above point, I do not see a clear motivation for this work. Also, I do not find the challenges of the problem addressed in this paper. Without knowing this, it is difficult to evaluate the contributions of this paper.

3. For the framework in Figure 2, I like the modeling Future component. This design is interesting and reasonable. However, I do not understand the modeling noise component. I fail to find its rationality in the paper. Figure 2 is the key to this paper. There is no description of that. The second paragraph of Section 3 unfortunately is not its description.

4. I lost the connection between Section 3.5 and Section 3.1 - 3.4. Since there are many variables, a notation table is suggested.

5. Stock prediction is a very interesting topic. However, I find the benchmark dataset used in this paper is out-of-the-date. This is an application-oriented paper, which should demonstrate its practicability. It is encouraged to see the performance of the proposed method on the recent stock prediction.

---

### Official Review · AnonReviewer3 · 2020-10-29
**Interesting work, Incremental and Unclear baselines**

**Rating:** 6
**Confidence:** 4

**Review:**

Summary:

This work extends the line of research for the problem of news-driven stock price movement prediction. In doing so,  the authors presents a Noisy Equity State (NES) representation model,
where they explicitly model equity state sequences by a recurrent state transition in an LSTM model. The proposed modeling approach also incorporates news impact and
noise impact employing attention and noise sampling techniques.

The proposed modeling approach offer the following advantages:
- incorporating additional inputs, such as past and present news, in equity state transition process
- addressing the effects of delayed news/insider trading and unknown factors by incorporating future news and random noise
- offers to filter and visualize the critical news based on the accumulated equity knowledge


Strengths:
- the paper is well motivated and clear
- extends news-driven stock price movement prediction task by introducing an LSMT based model, explicitly modeling equity state sequences and introducing additional signals/information. Also, recurrent equity representation based attention scores allow to filter out the non-critical news events.
- sound evaluation and analysis, including ablation and case studies
- extensive analysis to show the importance of past, present, future news and noise
- reports gains with the proposed approach


Weaknesses:
- the paper lacks novelty in terms of methodological contribution and is incremental. The work combines existing approaches and introduces additional signal/information
- unclear experimental setup of baselines: Do all the baselines models also use the same (additional) information as the proposed approach does.
- Table 3: marginal gains in accuracy (Compared to Ding et al 2016). Does the baseline Ding et al 2016 also use the same set of information for a fair comparison?
- For a fair comparison, the baselines (particularly Ding 2016) be trained on the same event embeddings as the proposed approach does, instead of using knowledge base embeddings. Additional investigation for fair comparison is required.

Questions:
- Why not BERT-based contextualized embeddings employed instead of ELMo?
- Table 3: Are the scores reported  after re-run of the baselines ?

---

### Decision · Program_Chairs · 2021-01-07
**Final Decision**

**Decision:**

Reject

**Comment:**

In this paper, the authors propose a model for integrating news representations for stock predictions. While the research direction has good value in real applications, it seems that this particular paper has not done a sufficiently good job in pushing the frontier of this direction. The reviewers have raised quite a few concerns, for example:
1)	Paper writing needs significant improvement (e.g., confusion regarding future news).
2)	Limited technical novelty as compared to previous works
3)	Benchmark datasets are out of date, baselines are a little weak and not well explained, more evaluation measures (Such as Sharpe value) are needed
4)	Marginal improvements over the baselines

The authors have not submitted their rebuttals. Therefore the concerns are still there and we do not think the paper is ready for publication at ICLR.